# Computational Fatigue Analysis of Auxetic Cellular Structures Made of SLM AlSi10Mg Alloy

**Miran Ulbin**[ID]**, Matej Borovinšek**[ID]**, Matej Vesenjak**[ID] **and Srečko Glodež \***

University of Maribor, Faculty of Mechanical Engineering, Smetanova 17, 2000 Maribor, Slovenia;
miran.ulbin@um.si (M.U.); matej.borovinsek@um.si (M.B.); matej.vesenjak@um.si (M.V.)
**\*** Correspondence: srecko.glodez@um.si; Tel.: +386-2-2207708

**Abstract:** In this study, a computational fatigue analysis of topology optimised auxetic cellular structures made of Selective Laser Melting (SLM) AlSi10Mg alloy is presented. Structures were selected from the Pareto front obtained by the multi-objective optimisation. Five structures with different negative Poisson's ratios were considered for the parametric numerical analysis, where the fillet radius of cellular struts has been chosen as a parameter. The fatigue life of the analysed structures was determined by the strain–life approach using the Universal Slope method, where the needed material parameters were determined according to the experimental results obtained by quasi-static unidirectional tensile tests. The obtained computational results have shown that generally less auxetic structures tend to have a better fatigue life expectancy. Furthermore, the fillet radius has a significant impact on fatigue life. In general, the fatigue life decreases for smaller fillet radiuses (less than 0.3 mm) as a consequence of the high-stress concentrations, and also for larger fillet radiuses (more than 0.6 mm) due to the moving of the plastic zone away from the edge of the cell connections. The obtained computational results serve as a basis for further investigation, which should be focused on the experimental testing of the fabricated auxetic cellular structures made of SLM AlSi10Mg alloy under cyclic loading conditions.

**Keywords:** aluminium alloy; SLM; auxetic structures; numerical analysis; fatigue

---

## 1. Introduction

Additive Manufacturing (AM) is a fabrication process that provides unique opportunities to manufacture customised parts with complex geometries or functionally graded materials [1,2]. As presented in [3,4], AM brings the promise of a new industrial revolution, offering the capability to manufacture parts through the repetitious deposition of material layers directly from a digital Computer-Aided Design (CAD) model. For those reasons, AM applies to a wide range of industries [5,6], including medical equipment [7,8], automotive [9,10], aerospace [11,12], and other consumer products [13]. Many AM technologies are being currently used in different engineering applications [14,15], e.g., Direct Metal Laser Sintering (DMLS), Selective Laser Melting (SLM), Electron Beam Melting (EBM), and Laser Metal Deposition (LMD). Among the AM processes, DMLS and SLM techniques are particularly widespread, especially for the aluminium alloy processing [16,17]. The DMLS technology is appropriate to build parts out of any metal alloy, while SLM can only be used with certain metals.

For AM technologies, Al–Si or Al–Si–Mg are the common alloys. In general, the static strength of AM parts depends on the density of the parts, as well as on the microstructure formed during the fabrication. As presented by Aboulkhair et al. [18] and Kempen et al. [19], the microstructure of the AM fabricated parts made of the AlSi10Mg alloy is usually finer (higher static strength) compared to the parts, which are fabricated via classical procedures (e.g., casting). On the other hand,

the microstructure of the AM fabricated parts is generally anisotropic and depends on the building and scanning directions during the AM process. For that reason, the monotonic material properties (yield stress, ultimate tensile strength) are also anisotropic and may strongly depend on the orientation of texture [20–22]. Zhou et al. [23] have shown that the microstructure, hardness, and static strength of the AM parts may be significantly influenced by additional heat treatment. Nevertheless, the fatigue behaviour of the AM parts is not well understood yet, although a lot of research was performed in the last decade [24]. Similar to the static mechanical properties, the fatigue strength of the AM parts primarily depends on their microstructure. However, the AM process usually leads to an increased surface roughness of the fabricated part, which decreases its fatigue strength [25]. Furthermore, material defects such as the porosity and insufficient layer bonding could result in an increased scatter of the fatigue properties [26]. Besides the fatigue analyses, the investigations on the fracture behaviour of the AM components have also been carried out, especially due to the high importance in the automotive industry and aerospace applications [27–29].

As mentioned above, AM technologies, including the SLM process, provide opportunities to manufacture very complex geometries or functionally graded materials, which are inaccessible through traditional manufacturing techniques. Typical lightweight structures, where the SLM process may be used, are pre-designed advanced cellular metamaterials and structures, which are difficult to fabricate by the traditional processes. In the last period, the research has been focusing on the mechanical response of the gradient lattice structures manufactured via the AM process in terms of energy absorption under the compressive loading conditions [30–32]. Some authors also investigated the fatigue behaviour of different AM lattice structures considering the effect of the cellular topology and geometrical imperfections on the fatigue life of the analysed structures [33–35]. The next group of metamaterials suitable to be produced using the AM technology are the auxetic cellular structures, which are the subject of this study (for a detailed description of the auxetic cellular structures, see Section 2.1).

The research work in the presented study is focused on the fatigue behaviour of five different auxetic cellular structures obtained by the topology optimisation and made of the SLM AlSi10Mg alloy, which is widely used in the aerospace and automotive industry because of its high specific strength, high corrosion resistance, and good flowability [36,37]. The fatigue life of different auxetic cellular geometries, also considering the influence of the fillet radiuses, is determined by the strain–life approach using Universal Slope method [38]. The material parameters were determined according to the experimental results previously obtained by the unidirectional quasi-static tests [39]. Numerical nonlinear finite element simulations were performed to determine the deformation behaviour and the stress and strain fields in the critical cross-sections of the analysed cellular structures. Then, the numerical results were used as the input parameters in the subsequent fatigue analyses.

## 2. Selection of Optimised Auxetic Cellular Structures

### 2.1. Auxetic Cellular Structures

Cellular structures have been increasingly used in modern engineering applications over the past decades due to their attractive combination of mechanical and thermal properties [40–42]. Recent advances in the AM technologies enable the fabrication of parts with complex internal cellular structure, which are commonly adjusted for specific applications. The auxetic cellular structures exhibit a negative Poisson's ratio and thus experience large volume changes under loading [43–45]. The auxetic structures tend to expand in the lateral direction when subjected to the uniaxial tensile loading and vice versa in the case of the compressive loading (Figure 1).

Such behaviour is made possible by their 2D or 3D designed hinge-like cellular skeleton with predefined geometry [46–48]. The auxetic cellular structures (made of metals [49], polymers [50,51], and textiles [52–54]) offer extraordinary mechanical properties (under quasi-static [55], impact [56,57], and fatigue [58,59] loading), i.e., variable stiffness [60], low density, large mechanical energy absorption

through deformation [61–63], and unique deformation behaviour [64,65]. Their cell shape changes rapidly through the complete auxetic structure under loading; thus, the loading efficiently spreads through the structure. The impact on a small part of the auxetic structure results in energy dissipation through its entire structure. With recent advances in the AM technologies [66], it is possible to fabricate the auxetic cellular structures with optimised and exactly predefined geometries (i.e., strut thickness, geometry, cell shape and size). The design and optimisation of auxetic structures can be done by using advanced Computer-Aided Engineering (CAE) methodologies to achieve and control the desired mechanical properties for specific applications [60,67]. Such precise fabrication control of the shape, size, and distribution of cells with the use of layered additive technologies makes the auxetic structures superior to the other conventionally manufactured cellular materials.

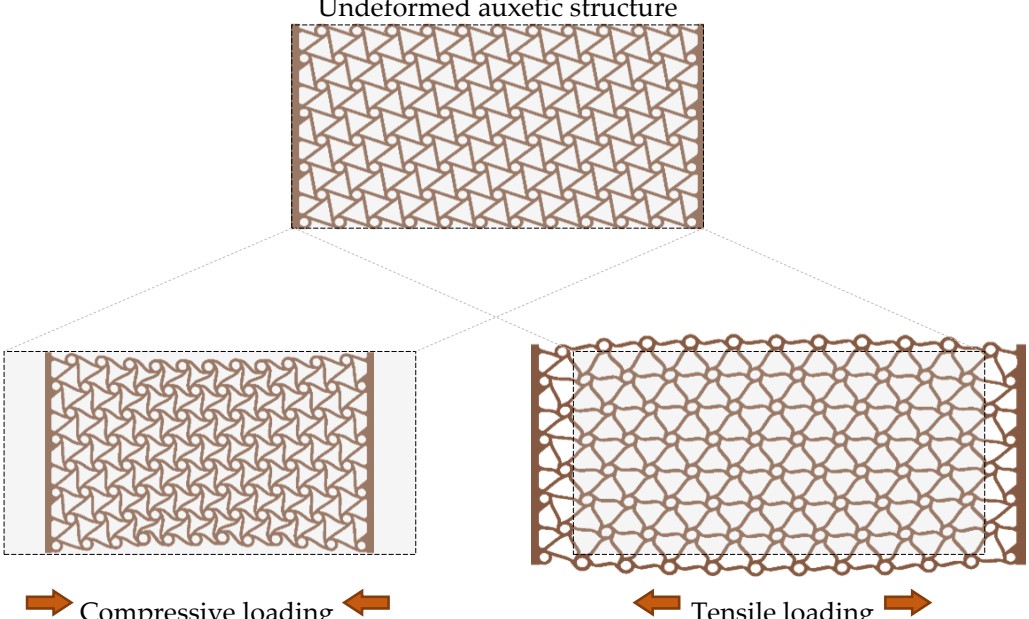

**Figure 1.** The behaviour of auxetic cellular structures under compressive and tensile loading.

### 2.2. Topology Optimised Auxetic Structures

Many auxetic cellular structures have been developed up to date [45]. Their shape, topology, and dimensions were usually custom-made for the task for which they were designed. Therefore, it is difficult to compare the fatigue properties between them. In the previous study by Borovinšek et al. [67], a procedure for the determination of optimal auxetic topology was developed using a numerical approach. The auxetic topology was modelled as a quarter cell of a 2D planar periodic structure. Appropriate boundary conditions were applied to account for the rest of the structure. The quarter cell model had a square shape, and its topology was represented by a grid of $10 \times 10$ equal squares (parameter regions), each of which could either represent material or a void region (Figure 2a). In this way, any topology of the quarter cell could be obtained by changing the content of the parameter regions.

The performance of the auxetic structure was regarded as a performance of the compliant mechanism, and as such, two objective functions were applied to determine the optimal topologies. The first objective function was used to measure the stiffness of the auxetic structure, while the second objective function measured its flexibility. Using two objective functions and a multi-objective optimisation procedure, which changed the values of the parameter regions, resulted in a set of best topologies called the Pareto front [67]. All solutions on the Pareto front are optimal and equally good. Thus, applying this procedure results in a set of topologies, which have an auxetic behaviour. Since they are optimised with the same procedure and with the same objectives, it is also possible to directly compare their fatigue behaviour.

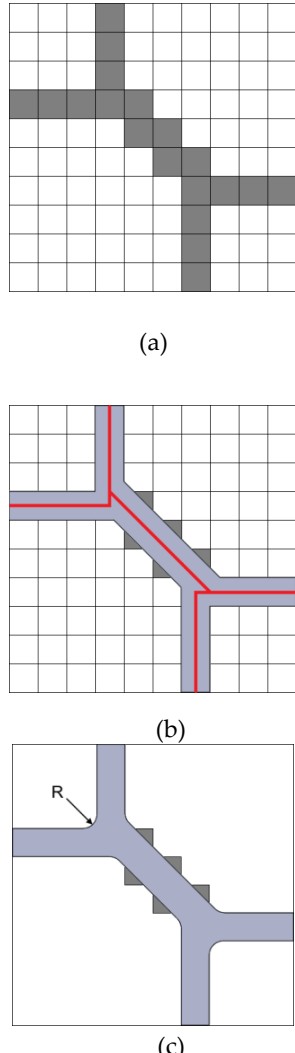

**Figure 2.** Base topology (**a**); medial axis topology (**b**); and rounded medial axis topology (**c**).

The drawback of this procedure is that the resulting optimal topologies of the quarter cell models are sharp-edged, since their topology consists of square-shaped parameter regions (Figure 2a). Sharp edges of the structure result in stress concentrations under load and do not allow a direct application of the obtained topologies from [67]. To study the fatigue behaviour of the obtained topologies, a geometry smoothing procedure was developed for this investigation. It was applied to reduce the number of possible stress concentration regions.

The smoothing procedure consists of the following five steps:

1.  Selection of the topology from the Pareto front (Figure 2a, dark grey area).
2.  Determination of the medial axis of the topology.
3.  Simplification of the medial axis to straight lines between the line connection points (Figure 2b, red lines).
4.  Thickening of the simplified medial axis to obtain the geometry; the horizontal and vertical lines retained their thickness, while a new thickness was defined to the oblique lines of the medial axis. The thickness of these lines was set to such a value that the topology area remained the same (Figure 2b, light grey area).
5.  All remaining sharp corners were rounded using an equal fillet radius R (Figure 2c).

From the Pareto front, obtained by the multi-objective optimisation, which contained 200 different topologies, five base topologies were selected for this study. The topologies of the Pareto front were

first sorted by their Poisson's ratio starting with the lowest one (Topology 1), followed by the median one (Topology 3), and so on until the highest one (Topology 5). An additional two topologies were selected between the lowest and the median Poisson's ratio (Topology 2) and between the median and the highest Poisson's ratio (Topology 4). The resulting five topologies are almost equally apart from each other on the Pareto front; thus, they are a good representation of the best solution set. The Poisson's ratio and the mass of all five base topologies are reported in Chapter 3. After selection, the smoothing procedure was applied, which resulted in five distinct quarter unit cell topologies (Figure 3). Their complete auxetic structures were made by mirroring and multiplying the quarter cell topology in such a way that the resulting periodic structures consisted of six-unit cells in both the (*x* and *y*) coordinate directions.

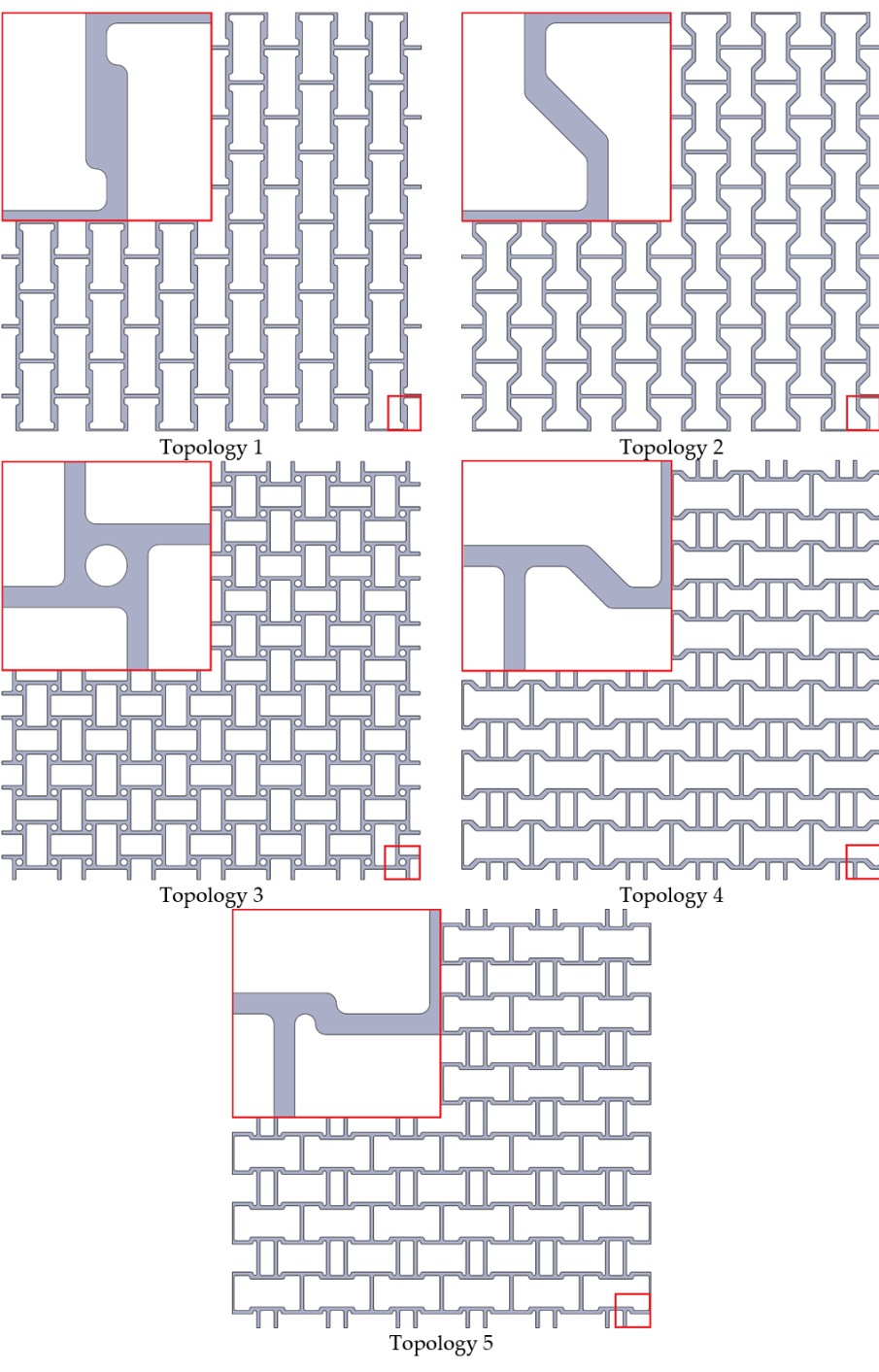

**Figure 3.** Selected quarter unit cell topologies and resulting auxetic structures.

## 3. Material Characterisation and Fatigue Analyses

### 3.1. Material Characterisation

The material used in this study is aluminium alloy AlSi10Mg. The specimens used for the further tensile tests have been fabricated by the SLM technology using the EOSINT-M-270 system without additional polishing. The general process data and the material composition are shown in Table 1. Figure 4a shows the geometry of the flat specimen, while the designated building and scanning fabrication directions are shown in Figure 4b.

**Table 1.** Material data sheet of Selective Laser Melting (SLM) AlSi10Mg alloy.

| Layer thickness: 30µm | | | | | | | | | | |
|---|---|---|---|---|---|---|---|---|---|---|
| Volume rate with the standard parameters: 4.8 mm$^2$/s | | | | | | | | | | |
| Surface roughness (as-built, cleaned): *Ra* 15–19 µm; *Rz* 96–115 µm | | | | | | | | | | |
| The density with the standard parameters: 2.68 g/cm$^3$ | | | | | | | | | | |
| Material composition (maximum values in %) | | | | | | | | | | |
| Al | Si | Fe | Cu | Mn | Mg | Ni | Zn | Pb | Sn | Ti |
| Bal. | 9–11 | 0.55 | 0.05 | 0.45 | 0.2–0.45 | 0.05 | 0.1 | 0.05 | 0.05 | 0.15 |

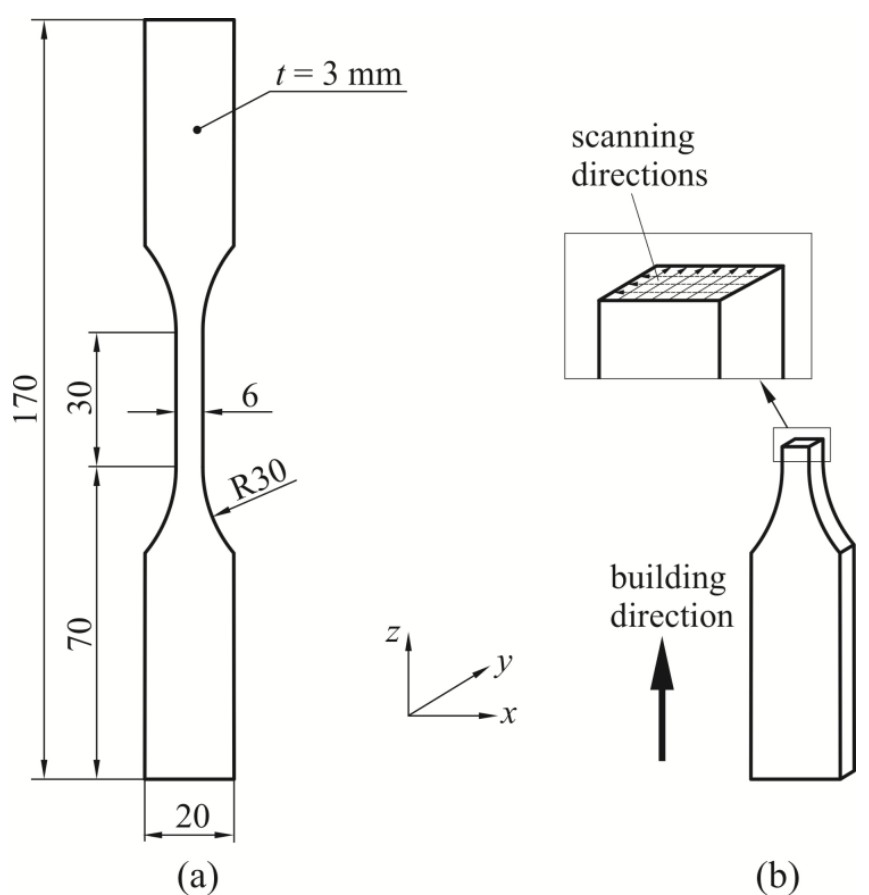

**Figure 4.** Geometry (**a**) and building direction (**b**) of a flat specimen. (all dimensions are in millimetres [mm]).

Figure 5 shows the polished surface of the fabricated specimen. The black circles represent the pores. They are significantly more frequent and larger near the specimen surface all around the circumference, up to 500 µm inwards. These near-surface pores were formed at the beginning/end of a scan in the *x*-direction and were not removed by the laser movement in the *y*-direction during the formation of the subsequent layer.

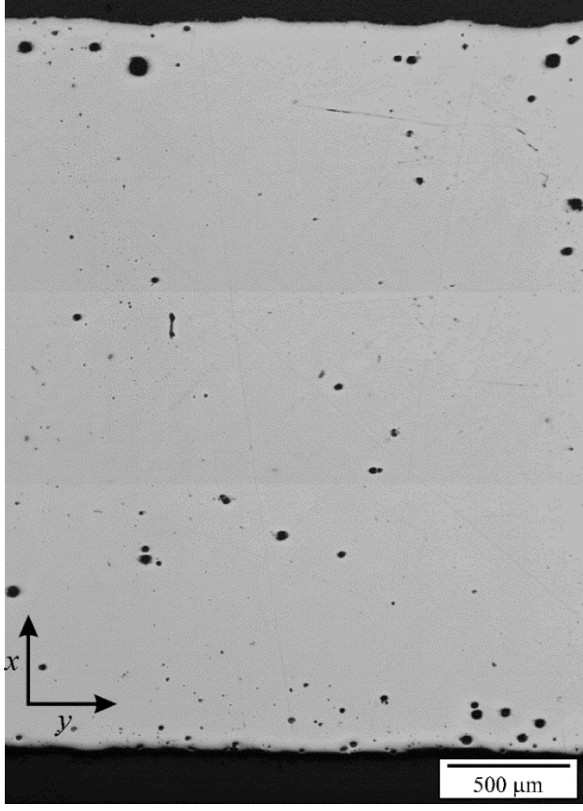

**Figure 5.** A light micrograph of the polished specimen [39].

Two specimens were used for the quasi-static tensile tests to measure the stress–strain (σ–ε) response. The tensile tests were carried out using the 100 kN MTS Landmark hydraulic machine (MTS Systems GmbH, Germany) at a room temperature of 23 °C. The tests were displacement controlled with a loading rate of 0.5 mm/min according to the DIN-ISO 6892 standard. The force was measured with the 100 kN MTS load cell, and the strains were measured with the MTS 834.11F-24 extensometer (MTS Systems GmbH, Germany). The measured quasi-static stress–strain responses [39] were almost the same for both specimens and are in a good agreement if compared to the experimental results from other researchers [68]. Based on the measured stress–strain responses (Figure 6), the average material parameters have been determined (Table 2).

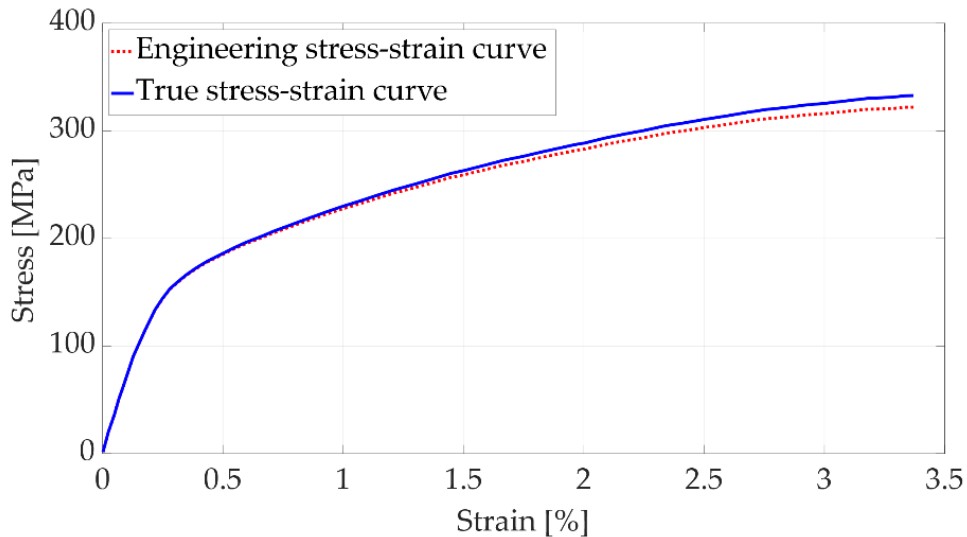

**Figure 6.** Stress–strain response of the SLM AlSi10Mg alloy.

| Elastic Modulus $E$ (MPa) | Poisson's Ratio $\nu$ (-) | Yield Stress $R_{p0,2}$ (MPa) | Ultimate Tensile Strength UTS $R_m$ (MPa) | Strain at Rupture $A_5$ (%) | True Fracture Strain $\varepsilon_f$ (-) |
|---|---|---|---|---|---|
| 70900 | 0.3 | 180 | 318 | 3.35 | 0.031 |

The true stress–strain ($\sigma_{true}$–$\varepsilon_{true}$) data were determined using engineering stress ($\sigma_{eng}$) and engineering strain ($\varepsilon_{eng}$) values as follows:

$$\varepsilon_{true} = \ln\left(1 + \varepsilon_{eng}\right) \tag{1}$$

$$\sigma_{true} = \sigma_{eng} \times \left(1 + \varepsilon_{eng}\right). \tag{2}$$

### 3.2. Strength Analysis

The numerical models of five auxetic structures were built in Ansys Mechanical [69], using a 2D plane stress formulation. The size of 10 mm × 10 mm was chosen for the one-unit cell, which results in a structure size of 60 mm × 60 mm and a nominal strut thickness of 0.5 mm. The finite element mesh consisted of plane stress parabolic quad finite elements PLANE183 with eight nodes. With the parametrical analysis, it was deduced that the results converge using a mesh, where the global element size is approximately 0.2 mm, with a consideration of curvatures, where the element size is even smaller. This resulted in numerical models with an average of 30,000 finite elements and 100,000 nodes. Each topology was constrained in the *y*-direction at the bottom and loaded with displacement at the top. All five structures were loaded with the same displacement of 0.4 mm in the + *y*-direction, thus creating a tension in the vertical direction. Such loading was chosen to induce plasticity in all structures. The higher loading displacement would cause severe plasticity in some structures. Similarly, a lower value of displacement would not cause plastic strain in some structures. The models were constrained in the *x*-direction in the middle of the structure to prevent horizontal movement. The boundary conditions are presented in Figure 7, where loading is acting in the vertical direction (*y*-direction).

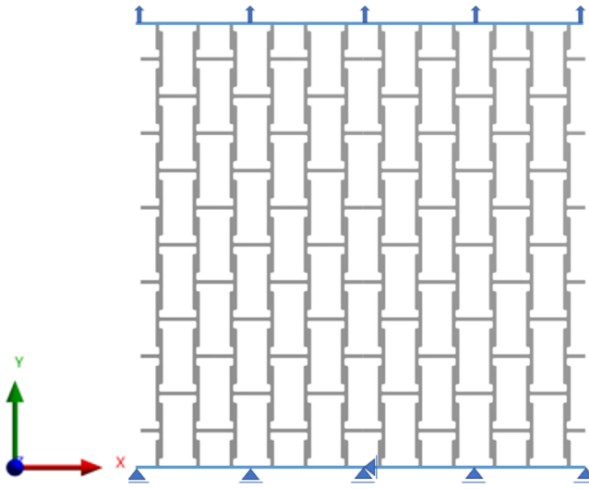

**Figure 7.** Boundary conditions.

The material in all presented numerical models was the SLM AlSi10Mg alloy with the properties presented in Section 3.1. The models were loaded with the displacement, which caused plastic deformation in all five models. Therefore, material nonlinearity was considered in the simulations. For plasticity, a multilinear isotropic hardening model was used, with the utilisation of the true stress–strain material data shown in Figure 6.

Numerical results in the form of total displacements are presented in Figure 8. The resulting displacement was scaled by the factor 11 for a better visualisation of all structures. It can be noted that they have different Poisson ratios. The Poisson ratios presented in Table 3 were calculated as the ratio between the longitudinal and lateral displacement. The Poisson ratios of the analysed structures are, to some extent, different than the unit cell's Poisson ratios. All the calculated Poisson ratios are negative, which proves that all the analysed structures are also auxetic. It can be seen that Topology 1 has the lowest Poisson ratio, while Topology 5 has the highest Poisson ratio. The Poisson ratios of the other three topologies are ascending between Topology 1 and Topology 5. Table 3 also reports the mass of each model showing that all structures have a similar weight. Only Topology 3 is a bit more massive.

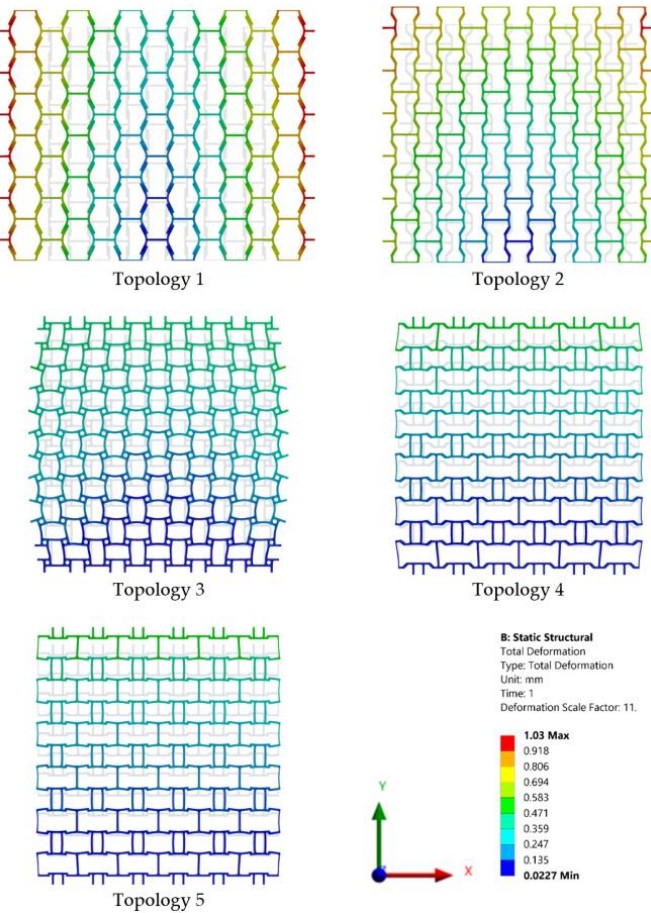

**Figure 8.** Deformed and undeformed structures.

**Table 3.** Properties and results of the analysed topologies.

| Topology | Mass (g) | Displacement x-direction (mm) | Equivalent Elastic Strain (mm/mm) | Equivalent Plastic Strain (mm/mm) | Poisson Ratio (-) | Unit Cell's Poisson Ratio (-) |
|---|---|---|---|---|---|---|
| 1 | 0.10 | 0.964 | 0.00408 | 0.0154 | −2.41 | −4.01 |
| 2 | 0.09 | 0.381 | 0.00307 | 0.0180 | −0.95 | −2.71 |
| 3 | 0.12 | 0.280 | 0.00362 | 0.0094 | −0.67 | −0.92 |
| 4 | 0.10 | 0.080 | 0.00312 | 0.0022 | −0.20 | −0.35 |
| 5 | 0.09 | 0.040 | 0.00323 | 0.0011 | −0.01 | −0.19 |

Figure 9 shows the location of the maximum plastic strain for each model. The area of the maximum plastic strain is also magnified to visualise the exact location of the developed plastic zones. Location of the maximum plastic strain is in all cases at the fillet radius of the auxetic structure.



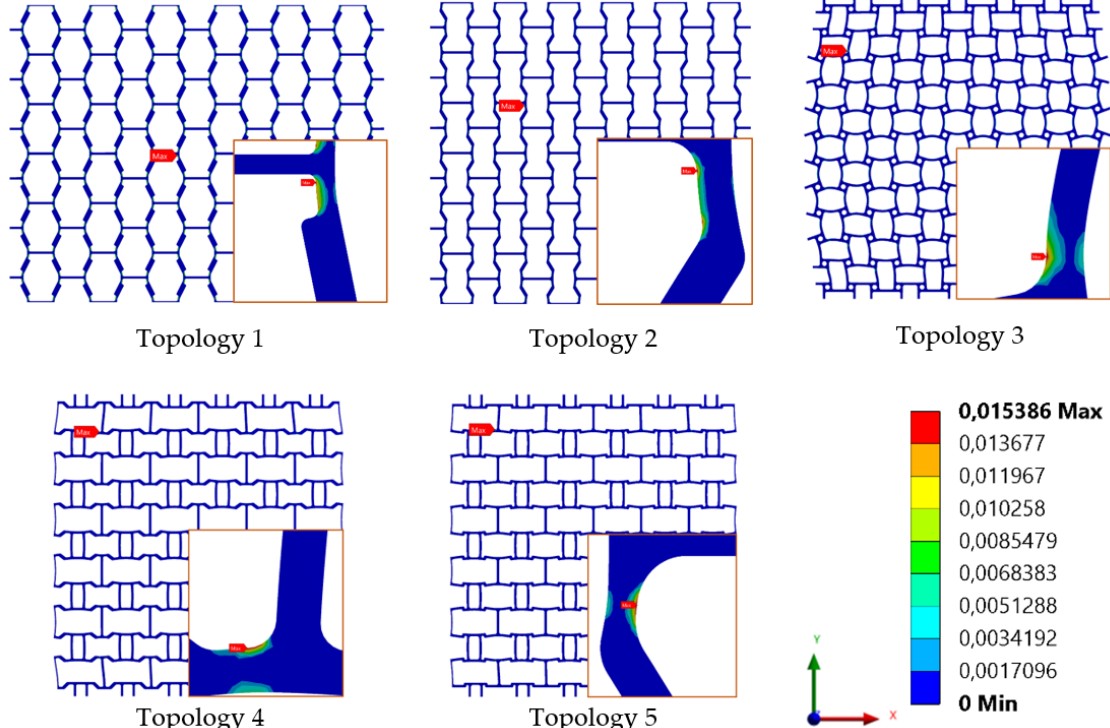

**Figure 9.** Location of the maximum equivalent plastic strain and its distribution.

Since the size of the radius is important, a series of parametric analyses were conducted with different radius sizes. Radiuses were scaled in a range of 0.1 and 0.9 mm with a step of 0.1 mm. Figure 3 shows the basic topology designs from which the numerical models were built. Most of the structures allow a wide range of radiuses, while Topology 1 and Topology 5 are limited because the two fillets intersect and prevent further radius scaling. Therefore, some radiuses were scaled, while the potentially intersecting radiuses were presented only up to the maximum possible values, e.g., Topologies 1 and 5. Each time the radius was scaled, the numerical model was regenerated with the same mesh parameters and the same boundary conditions as presented in Figure 7. In that way, it was possible to determine how radius sensitive the analysed structures are and the optimal radius for each structure in terms of fatigue.

Figure 10 shows the equivalent plastic strain for different fillet radiuses. For Topology 1, some fillets could not be scaled beyond 0.25 mm, while the remaining fillets are limited to 0.75 mm; therefore, the results for the radius of 0.9 mm for Topology 1 are missing.

From the results in Figure 10, it is clear that at a small fillet radius, the plastic strain occurs near the edge, which is placed where the crack initialisation is expected. With a larger fillet radius, the area of the plastic zone is wider, and the value of the equivalent plastic strain is lower in all structures. In most of the observed structures, this occurs at the radius of approximately 0.4 mm, with exceptions in Topology 1 (at the radius of approximately 0.2 mm) and Topology 4 (at the radius of approximately 0.3 mm). With further increase of the fillet radius, the value of the equivalent plastic strain increases. Furthermore, some new critical areas occur. Accordingly, large fillet radiuses create new critical places for the crack initiation, where sometimes the equivalent plastic strain has the same or larger value than observed in the case of very small fillet radiuses.

Then, the obtained numerical results were used for fatigue life determination. From each numerical result, the maximum equivalent elastic strain was added to the maximum equivalent plastic strain. This allows the calculation of the fatigue life cycle for each topology, including the fillet radius variations (Figure 10). For each structure, additional numerical analysis was performed to determine the yield load condition, when the structure behaves purely elastic.

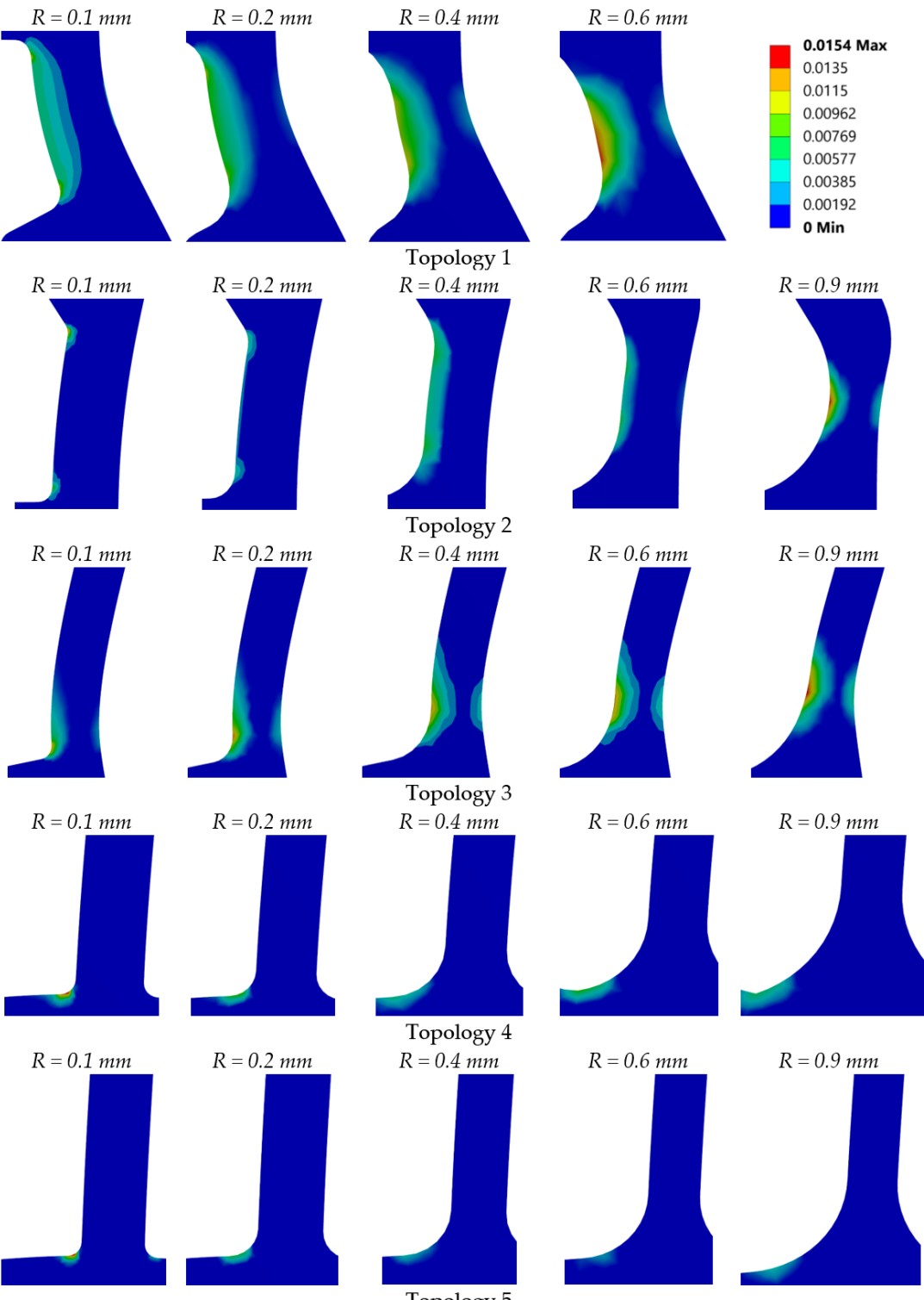

**Figure 10.** Equivalent plastic strain for different fillet radiuses.

### 3.3. Fatigue Life Determination

In the presented study, the strain–life approach is used to determine the fatigue life of treated cellular structures. In general, this approach is based on the knowledge of the stresses and strains that occur at locations, where the fatigue crack nucleation is likely to start, e.g., holes, fillets, grooves. In this approach, the fatigue behaviour of the material is described by the strain–life curve expressed by the Coffin–Manson relation between fatigue life $N_i$ and the total amplitude strain [70]. However, the material

low-cycle fatigue parameters should be determined previously using the appropriate low-cycle fatigue test. If it is not the case, an alternative method based on the material parameters obtained from a quasi-static test can be used. In this study, the method of Universal Slopes as proposed by Muralidharan and Manson [38] has been used to obtain the number of loading cycles $N_i$ required for the fatigue crack initiation in the critical cross-section of the treated cellular structure:

$$\varepsilon_{Ea} = 0.623\left(\frac{R_m}{E}\right)^{0.832}(2N_i)^{-0.09} + 0.0196\left(\varepsilon_f\right)^{0.155}\left(\frac{R_m}{E}\right)^{-0.53}(2N_i)^{-0.56} \tag{3}$$

where $\varepsilon_{Ea}$ is the equivalent amplitude strain, $E$ is the elasticity modulus, $R_m$ is the ultimate tensile strength, and $\varepsilon_f$ is the true fracture strain. Once the equivalent amplitude strain $\varepsilon_{Ea}$ is known, the number of stress cycles $N_i$ can be obtained iteratively from Equation (3) using Matlab software [71] and with a consideration of the material parameters $E$, $R_m$, and $\varepsilon_f$ given in Table 2. The results are shown in Figure 11.

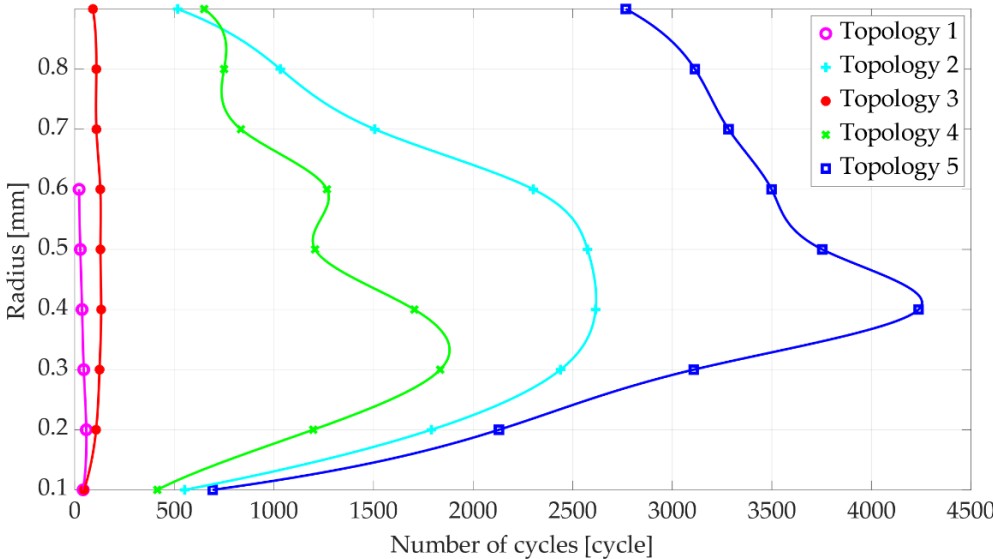

**Figure 11.** The fatigue life $N_i$ of different topologies and fillet radiuses.

For Topology 1, the expected fatigue life cycles are rather low, with a maximum value at the fillet radius of 0.2 mm. Topology 4 resulted in the best fatigue life expectance with the fillet radius of 0.3 mm. All other structures have the highest number of cycles with a radius of 0.4 mm, while the life expectancy of Topology 3 is very low. The best fatigue life is expected for Topology 5, which also has the highest Poisson ratio, making it the least auxetic structure presented in the paper. On the other hand, Topology 2 has the second-highest fatigue life expectancy, while having the second-lowest Poisson ratio. Therefore, Topology 2 could be used in cases where a highly auxetic and durable structure is needed.

## 4. Discussion

Five cellular structures were considered for the numerical analysis to identify the most suitable auxetic topology when considering the fatigue life under tension loading conditions. Structures were selected from the Pareto front obtained by the multi-objective optimisation. The structures were sorted by their Poisson ratio, starting with the lowest one (Topology 1) to the one that has the Poisson's ratio only slightly below zero (Topology 5), as can be seen from Table 3.

All five topologies were subjected to the same tensile loading conditions. The amount of the tensile load was high enough to cause plasticity in every topology but small enough that each topology would sustain several loading cycles until failure. The presented results of the numerical analysis show that generally less auxetic structures tend to have a better fatigue life expectancy.

The fillet radius has a significant impact on fatigue life. While it is expected that smaller radiuses would increase the probability of crack initiation, it is interesting to note that larger fillet radiuses also negatively influence the fatigue life. As the fillet radius increases, the plastic zone is moved away from the edge, as can be seen from Figure 10. Global tension loading displacement induces hinge-like behaviour in structure corners. While the hinges rotate, the structures struts are primarily loaded in bending. Since the struts have a constant cross-section area, the internal bending moment is equally distributed along their length, and the whole strut deforms to account for the hinge rotation. With increased fillet radius, the strut length is decreased. To account for the same hinge rotation, a shorter strut must deform more, which causes a higher rate of plasticity with the same tension loading displacement.

Topology 1 has the lowest (most negative) Poisson ratio, which makes it the most auxetic structure analysed herein. Computed displacements in the lateral direction ($x$-direction) are more than two times larger than the loading displacements in the longitudinal direction ($y$-direction). However, the prescribed tension displacement is quite large for this structure, as it results in an equivalent plastic strain of 1.5%. The fatigue life under such loading conditions would be very short, which is also clearly shown in Figure 11. Loading would have to be reduced at least 10 times to achieve the pure elastic behaviour of the structure, which questions its usability compared to the other analysed structures.

As can be seen from Figure 11, Topology 2 is the second most auxetic structure and has the second-best fatigue life. This makes Topology 2 the best choice for an application where the auxetic response is needed. With the Poisson ratio of −0.95, the displacement in the lateral direction ($x$-direction) is almost identical to the tension loading displacement. With the applied loading, the structure would sustain approximately 2500 cycles, and if the loading displacement would be decreased to less than half of the applied loading, the structure would behave purely elastically, while still exhibiting the same displacement in both directions. This makes Topology 2 the most appropriate choice for the auxetic structure.

By analysing Topology 3, it can be observed that the lateral displacement is nearly half the size of the tension loading displacement. However, the equivalent plastic strain is approximately 1%, and the structure has the second-worst fatigue life expectancy Additionally, Topology 3 is the most massive structure compared to the other topologies. Therefore, this structure is not recommended for an application subjected to cyclic loading unless a specific need utilising such a Poisson ratio is required.

The Poisson ratio of Topology 4 is equal to −0.2. Topology 4 also has a high expected fatigue life. Similar to Topology 2, this structure would behave purely elastically when the loading displacement is reduced by half. Comparing Poisson ratio, it is evident that the lateral displacement would be five times lower, as it was observed in the case of Topology 2. It could be recommended for applications where a small amount of auxetic behaviour is required. Nevertheless, Topology 2 shows a better performance in terms of lateral displacement and fatigue life.

Topology 5 shows the smallest auxetic effect, as its Poisson ratio is close to zero. At the same time, it is the structure with the highest number of fatigue life cycles determined, as can be seen from Figure 11. If the tension loading displacement would be reduced only by 25%, the structure would behave purely elastic. Therefore, Topology 5 could be used for applications where the transversal strain should be limited.

## 5. Conclusions

In this study, the fatigue behaviour of auxetic cellular structures obtained by multi-objective topology optimisation and made of SLM AlSi10Mg alloy is presented. Five structures with different Poisson ratios were considered for the parametric numerical analysis, where the fillet radius of cellular struts has been chosen as a parameter. The fatigue life of the analysed structures was determined by the strain–life approach using the Universal Slope method, where the needed material parameters were

determined according to the experimental results obtained by the quasi-static unidirectional tensile tests. Based on the fractography of the quasi-static specimens, numerical simulations, and fatigue analyses, the following conclusions can be made:

- The microstructure of treated SLM AlSi10Mg alloy consists of bands, which were produced by varying the scan direction in each subsequent layer. The material porosity is more frequent and larger in the near-surface layers, where pores were formed at the end/beginning of scanning during the SLM process.
- The obtained material parameters by the quasi-static tests (proportional limit, ultimate tensile strength, strain at rupture) are comparable to the results presented by the other researchers.
- The obtained computational results have shown that less auxetic structures (higher Poisson ratios) tend to have a better fatigue life expectancy (the longest fatigue life has been obtained for the structure topology with the minimum auxetic characteristics).
- The fillet radius of cellular struts has a significant impact on fatigue life. Computational analyses have shown that the fatigue life decreases for smaller fillet radiuses (less than 0.3 mm) as a consequence of high-stress concentrations and also for larger fillet radiuses (more than 0.6 mm) due to the movement of the plastic zone away from the edge of the cell connections. Besides the fillet radius sizes, other reasons for stress concentrations might exist. This should be investigated in future research.
- The fatigue life in this study was obtained using the simplified Universal Slope method, where the material parameters were determined by quasi-static tensile tests considering the partly porous structure of analysed SLM AlSi10Mg alloy (Figure 5). For a more accurate determination of fatigue life of real components made of SLM AlSi10Mg alloy, the comprehensive Low Cycle Fatigue (LCF) tests should be performed to determine the appropriate LCF material properties, where the influence of pores will be considered in detail. Once the LCF material parameters are known, the standardised strain life approach could be used for the subsequent fatigue analysis.
- The obtained computational results provide a basis for further investigation including experimental testing of the fabricated auxetic cellular structures made of SLM AlSi10Mg alloy under the cyclic loading conditions.

**Author Contributions:** Conceptualization, M.U. and S.G.; investigation, S.G.; methodology, M.V.; project administration, M.V.; software, M.B.; Writing—review and editing, M.U., M.B., M.V. and S.G. All authors have read and agreed to the published version of the manuscript.

**Funding:** This research was funded by the Slovenian Research Agency—ARRS, basic research project No. J2-8186 and research core No. P2-0063.

**Conflicts of Interest:** The authors declare no conflict of interest.

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
