# Peer review of "Computational Fatigue Analysis of Auxetic Cellular Structures Made of SLM AlSi10Mg Alloy"

_metals, doi:10.3390/met10070945_

Round 1

Reviewer 1 Report

The topic of the paper is very interesting.

However, the paper regards only a numerical approach, while the title indicates a completely different work: in fact, the fatigue life of lattice structures produced by SLM requires experimental testing, while the manuscript is oriented only to numerical approach, even without any specific background.

The experimental section (n.3- change the title, please) does not contain any significant informations about the process parameters during SLM. In particular, the process parameters can be selected for printing massive or lattice structure: which is the case?

Figure 5 shows a quite bad sample. The authors should check the residual porosity. It would be better to show the cross section of the cellular structure.

Once more time, the experimental data of fatigue tests is missing at all.

Author Response

Dear Editor.

The Reviewers’ comments regarding our original submitted manuscript are very welcome. We have tried our very best to satisfy all Reviewers’ recommendations, and we sincerely hope that the revised manuscript will now be satisfactory. All changes regarding the original manuscript are marked in blue in the revised manuscript.

The answers on the reviewer's comments can be found in the attached file "Metals-831405-Response to Reviewers' comments_R1".

Reviewer 2 Report

The present research deals with the study of the fatigue behaviour of auxetic topology optimised structures of AlSi10Mg alloy made by Selective Laser Melting process. The fatigue performances were evaluated considering the effect of the fillet radius of the auxetic structure by using the strain-life approach.

The paper is interesting but for improving the manuscript, it is advisable to address the following comments:

  • Even if the motivation for optimising the topology of auxetic structures is clear, the explanation of the smoothing procedure quoted at line 125 requires to be improved: the authors can include a bullet list of the steps of the smoothing or a figure representing the algorithm that includes each image reported in figure 2.
  • It would increase the soundness of the research to include in the text the Pareto front and more in-depth on its application to the present study.
  • Please provide more explanations on the choice of the Topologies from Pareto’s front.
  • Please improve the quality of Figure 4b.
  • Table II: the author said that two samples were addressed to static tensile tests. So, the values presented in Table II represent the mean values through two tests. Please provide specifications. In order to have a statistical significant results it would have been better to increase the sample number being tested.
  • The main concern of the present paper is such that the work for to the fatigue life assessments refers mostly to the numerical analysis. In effect, the authors used the equivalent amplitude strains assessed by means of the numerical analysis to obtain the number of cycles to failure. Please clarify in the text the reason for a purely numerical investigation.
  • Fatigue life usually is discussed referring to the Strain/N graph. Why in figure 11 the axes are inverted, is there any particular reason for that representation?
  • Can the author motivate the choice of the applied load (displacement of 0.4 mm) ? Which was the strain ratio applied? Did the author consider the effect of strain-ratio (minimum strain vs maximum strain) on the fatigue behaviour?

Author Response

(The authors gave the same response as above.)

Reviewer 3 Report

General

The authors deal with the question of the fatigue behavior of auxetic structures, which is a relevant for additive manufacturing. For this purpose, a numerical model is analyzed for five auxetic structures in different states. The paper makes a valuable contribution in the field of additive fabrication of auxetic structures and contributes to a further scientific development of this topic. The introduction and the problem are described in a comprehensible way and references are given to an appropriate extent and actuality.

The reviewer suggests to use the term "Laser Powder Bed Fusion" (LPBF/L-PBF) as a general term in the contribution for describing the process - instead of SLM. For example, in 3.1 the use of a EOSINT M270 is mentioned, however, EOS itself refers to their metal processes as DMLS and not SLM. SLM Solutions on the other hand registered "SLM" for their process. The reviewer is of the opinion that any commercially protected terms should be avoided to ensure clarity.

The reviewer finds the title (and first sentence in the Abstract) somewhat misleading, as it does not indicate that it is essentially a numerical analysis. The reader might come to the conclusion that auxetic structures are physically produced and analysed accordingly. It is suggested to clarify this by including "numerical analysis" already in the title and in the beginning of the abstract.

The authors are encouraged to check if graphics with a higher resolution can be used, some of them seem a bit pixelated.

Introduction

The introduction is reasonably comprehensive and offers the reader a good insight into the problem and state of the science.

Selection of Optimized Auxetic Cellular Structure

Figure 1 gives the reader a good and compact insight into the behavior of auxetic structures. Reference is made to the possibilities of additive manufacturing and the fact that there is a high degree of freedom in the design of cellular structures which cannot be realized otherwise.

My understanding is that the topology optimized structures originate from Ref. 65. However, under 2.2. paragraph 3: The wording "The drawback of this procedure" in connection with "...the following smoothing procedure was applied..." implies that here original scientific investigations follow which are based on a given state of the science: Please clarify whether these are completely results from reference 65 (Borovinsek) or whether the execution of the smoothing procedure is an original component of the investigations for this publication. Against this background, it should also be more clearly stated whether topology 1-5 originates from Ref. 65 or were developed in the context of this publication.

Fatigue Analysis

Table 1 should, according to the text, include general process data. However, besides a minimum layer thickness, no process data is included. In my understanding, this would be e.g. scan speed, hatch distance, platform temperature, laser powder etc. or at least volume energy density. Surface roughness and density are process outcomes. Additional question with respect to "minimum layer thickness": this wording implies a variation in layer thickness towards a larger thickness for the experiments. But this does not seem to be the case, please leave out "minimum".

Please provide information, how the polishing of your flat tensile specimen was done and how the surface roughness is after polishing (e.g. add information to Table 1).

Figure 5 provides a micrograph and the occurence of pores are mentioned in the text. The authors do not use or discuss the occurrence of the pores further. This is necessary, however, because pores in laser-additive manufactured components have a significant influence on fatigue behavior, especially in dynamic conditions, which is the main topic of the paper. If the occurrence of pores in the real component cannot be considered in the numerical model due to its complexity, this should be mentioned and also stated in the discussion as a possible source of error when specifying  fatigue cycles. Here your approach, the use of the Universal Slopes, should be mentioned and briefly described. So, the assumption is, pores are in the part but their impact in dynamic conditions is considered within the Universal Slope method? This should also be stated early on.

Line 198/199: Would you please explain why there is a scaling factor of 11. Is this for an improved visualization in Fig. 8?

Discussion

The authors' finding that small but also large radii have a negative influence is indeed interesting. The authors name a general application scenario for each topology-optimized unit cell, which is a valuable aid for scientific investigations based on this for an application-specific unit cell design.

The discussion is overall comprehensive and addresses the essential aspects. However, it would also be desirable to consider possible sources of error in the numerical analysis. Here I am coming back to the pores and the Universal Slope method. How valid is this approach and in consequence how valid are the specifically mentioned life cycles? Your analysis is based on the mechanics of two flat tensile specimen - how do the individual results of those specimen differ? How pronounced is their standard deviation, are those results a valid basis to work with?

Conclusion

line 342: Here and with reference to Fig. 5 you mention pores, but you do not consider or discuss them. This should definitely be addressed.

line 346: Please add a reference in 3.1 / Table 2 to results of other researchers if you mention the comparability of your results.

Author Response

(The authors gave the same response as above.)

Reviewer 4 Report

Dear Authors,

The paper entitled „ Fatigue behaviour of topology optimised auxetic 2 cellular structures made of SLM AlSi10Mg alloy” the optimisation of the auxetic cellular structures is presented. The paper consist of five sections and 70 references. In the first two paragraphs the introduction of the problem is presented. Five different geometries are investigated with respect to the static and fatigue loading conditions (section 3). In section 4 obtained results are discussed and sumarized and in the last section conclusions are given. The performed finite element analyses revealed the significant influence of the cell geometry and shape of the notches on the fatigue strength/life. Finally, on the basis of the numerical analyses and assumed fatigue model the Authors proposed optimal structure of the auxetic cellular structure.

Major Remarks:

  1. The title of the section 3 is misleading. In the first two subsections there is no experimental fatigue tests as well as no numerical fatigue analyses. Fatigue appears in subsection 3.3. In my opinion this section should be divided to the two sections with I. static analyses - Material characterization (old section 3.1) and strength analysis (old section 3.2) and II. Fatigue life determination (old section 3.3).

  2. The performed analyses are made with optimiastion of the cell geometries with various radius of the notches. In my opinion the term „topology optimization” is inappropriate. The topology optimisation is the special method of optimisation (see Refs [1-3]) and is not used in the proposed paper. I propose to use „parametric optimisation” instead „topology optimization” in the whole paper as well as in the title of the paper.

  1. Figure 6 : How the true stress – strain curve was determined? Please provide more information about that.

  2. Figures 7—9 : The local coordinate system is missing or too small. Including loading direction in figures 8-9 would make reading easier.

  3. Some shapes of the cells are similar to the cut-outs used in the pressure vessels. Some procedures of optimization of such cut-outs are discussed in Ref [3]. It can be observed that not only radius have important influence on stress concentration.

  4. Line 245: ”For each structure, additional numerical analysis was performed to determine the yield load condition, when structure behaves purely elastic and therefore has an infinite fatigue life.„ I am not sure about this assumption. In example in alloy steels fatigue may occur under purely elastic stresses. The fatigue limits are generally lower the yield limits. It means that purely elastic response of material do not means infinity fatigue life. Please revise.

  5. The fatigue model used in the analyses have 7 different coefficients: 0.623, 0.832,-0.09, 0.0196, 0.155, -0.53, -0.56. How these values have been achieved if the Authors performed only unixial tensile test?

Minor remarks

  1. Line 73: „The material parameters were determined according to the experimental results previously obtained by the unidirectional tests„ -static or fatigue? It should be given here.

  2. Line 139: Table 3 is cited before citation of the table 2.

  3. I propose to add references in the following lines:

    - line 72 „....using Universal Slope method.” - reference to this method,

    - line 103: „Many auxetic cellular structures have been developed up-to-date.„ - please provide some references of such structures.

    -line 118 „... topologies called the Pareto front.„ - please give more information or references reffering to the Pareto front

  4. Line 178”The numerical models of five auxetic structures were built in Ansys Mechanical [67], using a 2D plane stress formulation „ Please provide a type element used in the FE analyses.

  5. Lines 200-203 ”Poisson’s ratios were calculated as the ratio between the longitudinal and lateral displacement. The calculated Poisson’s ratios and other results are given in Table 3. The Poisson’s ratios of the analysed structures are to some extent different than unit cell’s Poisson's ratio as can be seen from Table 3. „ - Please revise grammar and style

[1] Suzuki, K., Kikuchi, N. A homogenization method for shape and topology optimization. Computers Methods in Applied Mechanics and Engineering 1991, 93, 291-318.

[2] Sleesongsom, S.; Bureerat, S. Topology Optimisation Using MPBILs and Multi-Grid Ground Element. Appl. Sci. 2018, 8, 271.

[3] Szybiński, B., Romanowicz, P.J, Optimization of flat ends in pressure vessels. Materials 2019, 12(24), 4194; https://doi.org/10.3390/ma12244194

Kind Regards,

Author Response

(The authors gave the same response as above.)

Round 2

Reviewer 1 Report

Dear authors, the paper has been improved and now for me it can be accepted after the following modification: the process is not DMLS bu SLM. Please, change the title and the other parts.

Author Response

Comment #1:

Dear authors, the paper has been improved and now for me it can be accepted after the following modification: the process is not DMLS bu SLM. Please, change the title and the other parts.

Response:

The DMLS was replaced with the SLM.

Reviewer 4 Report

Revision 2:

Remarks:

  1. The true stress-strain curve is determinated with the use of the classical approach for typical engineering materials while the investigated material in the paper reveal unusual behaviour caused by auxetic cellular structure.

  2. The paper is referenced to the fatigue analyses of auxetic cellular structure and such fatigue analyses may be completely wrong due to the not calibrated fatigue model for investigated material. Moreover, there is no experimental fatigue analyses which can prove or verify applied fatigue model and optimization technique.

    Substantiation :

    The fatigue model, which was assumed in the fatigue analyses is proposed as universal model, in which coefficients (named by Authors as constants) have been determined based on typical and popular 47 materials, including steels and aluminum and titanium alloys. Each of the parameters was determined using a least squares analysis for such materials. However, the investigated material in the proposed paper significantly differs from materials for which such fatigue model was proposed and calibrated. The following differences can be pointed: manufacturing technology (Direct Metal Laser Sintering), material (AlSi10Mg), structure (pores, cellular), Poisson's ratio (negative) and anisotropy.

Author Response

Comment #1:

The true stress-strain curve is determinated with the use of the classical approach for typical engineering materials while the investigated material in the paper reveal unusual behaviour caused by auxetic cellular structure.

Response:

The SLM AlSi10Mg alloy is the base material of the analysed structures, but it is otherwise typical engineering material. Auxetic behaviour is property of the structures, while the base material Poisson’s ratio was 0.3. A column with the Poisson’s ratio of the base material was inserted into Table 2.

Comment #2:

The paper is referenced to the fatigue analyses of auxetic cellular structure and such fatigue analyses may be completely wrong due to the not calibrated fatigue model for investigated material. Moreover, there is no experimental fatigue analyses which can prove or verify applied fatigue model and optimization technique.

Substantiation :

The fatigue model, which was assumed in the fatigue analyses is proposed as universal model, in which coefficients (named by Authors as constants) have been determined based on typical and popular 47 materials, including steels and aluminum and titanium alloys. Each of the parameters was determined using a least squares analysis for such materials. However, the investigated material in the proposed paper significantly differs from materials for which such fatigue model was proposed and calibrated. The following differences can be pointed: manufacturing technology (Direct Metal Laser Sintering), material (AlSi10Mg), structure (pores, cellular), Poisson's ratio (negative) and anisotropy.

Response:

Authors agree with the reviewer, that the Eq. (3) related to the method of Universal Slopes is based on the previous fatigue analyses made on 47 materials, including steels and aluminium and titanium alloys. Here, we would like to point out that the mechanical properties of analysed SLM AlSi10Mg alloy are very similar if compare to classical AlSi10Mg alloy and, consequently, we used the above equation when determining the fatigue life. On the other hand, some characteristics considered in the numerical simulations (porosity, negative Poisson’s ratio) are typical parameters of analysed cellular structures and influence on the strain and stress field in the critical cross-section. Once the strain field is obtained, the fatigue life can be determined using Eq. (3) where only the mechanical properties of the base material are considered.  

Round 3

Reviewer 4 Report

Dear Authors,

I don't have any remarks.

Best Regards,